# Isn't high school bad enough already? Rates of gender harassment and institutional betrayal in high school and their association with trauma-related symptoms

**Monika N. Lind** **\*, Alexis A. Adams-Clark, Jennifer J. Freyd**

Department of Psychology, University of Oregon, Eugene, Oregon, United States of America

\* mlind2@uoregon.edu

**Data Availability Statement:** All data files are available on figshare (DOI: https://doi.org/10.6084/m9.figshare.11551218.v1).

## Abstract

Germinal studies have described the prevalence of sex-based harassment in high schools and its associations with adverse outcomes in adolescents. Studies have focused on students, with little attention given to the actions of high schools themselves. Though journalists responded to the #MeToo movement by reporting on schools' betrayal of students who report misconduct, this topic remains understudied by researchers. Gender harassment is characterized by sexist remarks, sexually crude or offensive behavior, gender policing, work-family policing, and infantilization. Institutional betrayal is characterized by the failure of an institution, such as a school, to protect individuals dependent on the institution. We investigated high school gender harassment and institutional betrayal reported retrospectively by 535 current undergraduates. Our primary aim was to investigate whether institutional betrayal moderates the relationship between high school gender harassment and current trauma symptoms. In our pre-registered hypotheses (https://osf.io/3ds8k), we predicted that (1) high school gender harassment would be associated with more current trauma symptoms and (2) institutional betrayal would moderate this relationship such that high levels of institutional betrayal would be associated with a stronger association between high school gender harassment and current trauma symptoms. Consistent with our first hypothesis, high school gender harassment significantly predicted college trauma-related symptoms. An equation that included participant gender, race, age, high school gender harassment, institutional betrayal, and the interaction of gender harassment and institutional betrayal also significantly predicted trauma-related symptoms. Contrary to our second hypothesis, the interaction term was non-significant. However, institutional betrayal predicted unique variance in current trauma symptoms above and beyond the other variables. These findings indicate that both high school gender harassment and high school institutional betrayal are independently associated with trauma symptoms, suggesting that intervention should target both phenomena.

**Funding:** The authors received no specific funding for this work.

**Competing interests:** The authors have read the journal's policy and have the following competing interests: Jennifer J. Freyd, PhD, is the Founder and President of the Center for Institutional Courage, Professor of Psychology at the University of Oregon, and Faculty Affiliate of the VMware Women's Leadership Innovation Lab at Stanford University. She is also a Member of the Advisory Committee, 2019-2023, for the Action Collaborative on Preventing Sexual Harassment in Higher Education, National Academies of Science, Engineering, and Medicine. Freyd is the author of the Harvard Press book Betrayal Trauma: The Logic of Forgetting Childhood Abuse. Her most recent book Blind to Betrayal, co-authored with Pamela J. Birrell, was published by John Wiley. For these books Freyd receives royalties. She is often paid honoraria for presentations and she has served as a consultant on some legal cases and for some governmental and non-profit organizations. There are no patents, products in development or marketed products associated with this research to declare. This does not alter our adherence to PLOS ONE policies on sharing data and materials.

# Introduction

Important work occurs in adolescence. In the midst of physical, neurological, and psychological changes, adolescents undertake the intertwined tasks of social learning and identity development. Adolescents must also develop future-oriented, goal-directed skills that allow them to assume adult roles. Adolescence is a sensitive period, second only to infancy in its plasticity and its potential for positive or negative inflection points [1]. In many ways high school is the "workplace" where the important work of adolescence occurs. This study endeavors to investigate whether gender harassment and institutional betrayal create a hostile work environment in high schools and interfere with healthy adolescent development.

Gender harassment, a subdomain of sex-based harassment, includes verbal and non-verbal expressions and actions that derogate a person or group based on their gender [2]. Importantly, gender harassment does not necessarily convey sexual interest to the harassed party; rather, gender harassment conveys derision and/or hostility [2]. Leskinen and Cortina captured gender harassment in five domains: sexist remarks, sexually crude or offensive behavior, gender policing, work-family policing, and infantilization [3].

Most of the research on gender harassment has focused on its prevalence among adults, especially in the workplace. In adult samples, experiencing gender harassment is associated with a host of negative outcomes, including decreased psychological and professional well-being [2]. Much less is known about gender harassment in adolescence. Landmark studies by the AAUW capture aspects of gender harassment in their broader measure of sex-based harassment, and the data suggest high prevalence of gender harassment for boys and girls [4]. Like most other studies of sex-based harassment in high school, however, the AAUW studies assign only a few survey items to gender harassment and they do not test gender harassment as its own construct in their models. Gender harassment in high school deserves more attention, with parallel aims of describing its prevalence and dimensions and investigating its relationship with negative outcomes.

Trauma-related symptoms comprise one such category of potential negative outcomes that merit investigation. Trauma-related symptoms include dissociation, somatic complaints, mood-related problems, sleep disturbance, and others [5]. Sex-based harassment is associated with trauma-related symptoms in adults and adolescents [6, 7]. Evidence suggests that in adults, frequent gender harassment is as strongly associated with trauma-related symptoms as occasional, more coercive sex-based harassment [8]. It stands to reason that frequent gender harassment in high school could have similar associations with increased trauma-related symptoms.

Another limitation of the existing literature on high school sex-based harassment is that it does not address the high school institution's response to the harassment. Though journalists contributed to the #MeToo movement by publishing widely on the ways in which schools betray students who report misconduct [9, 10], this topic remains understudied by researchers. Institutional betrayal is characterized by the failure of an institution, such as a school, to protect individuals dependent on the institution [11]. Germinal studies have documented the occurrence of institutional betrayal across diverse contexts, including health care, the military, and college campuses [12–14]. These studies suggest that institutional betrayal may exacerbate the deleterious effects of stressful events.

Although these studies have provided foundational results regarding the prevalence and potential consequences of institutional betrayal, they have conceptualized institutional betrayal as only occurring in the aftermath of a traumatic event. This manifests in the studies' designs in that only participants who reported experiencing a traumatic event were asked to report on their experience of institutional betrayal. Evidence of negative outcomes related to witnessing

gender harassment and the normalization of sex-based harassment by schools suggests an alternative conceptualization of institutional betrayal [15, 16]. As the foundational literature indicates, institutional betrayal can be incident-specific, but it also may occur at the climate level.

The prevalence and dimensions of gender harassment in high school deserve detailed investigation, and the experience of gender harassment may be connected with trauma-related symptoms. Furthermore, high schools as institutions may play an important role in the association between harassment and trauma-related symptoms. For these reasons, we conducted an empirical study investigating the rates and correlates of high school gender harassment and institutional betrayal reported retrospectively by current undergraduates.

Our primary aims were (1) to describe the rates of gender harassment and institutional betrayal in high schools and (2) to investigate whether institutional betrayal moderates the relationship between high school gender harassment and current trauma symptoms. In our pre-registered hypotheses, we predicted that high school gender harassment would positively predict current trauma symptoms and that institutional betrayal would moderate this relationship.

## Methods

### Participants

The Office of Research Compliance (Institutional Review Board) at the University of Oregon approved this study. Informed consent was obtained in writing. Participants were recruited over the course of two academic terms from an undergraduate Human Subjects Pool managed by a public university located in the northwest United States. The institution's Human Subjects Pool consists of undergraduate students over the age of 18 from psychology and linguistics courses. Students receive course credit for completing studies, and they are permitted to end their study participation at any time. Because of the design of the Human Subjects Pool, participants are not aware of the topic of the research prior to signing up to participate, which protects against biased self-selection [17]. This process was approved by the institution's Office of Research Compliance (Institutional Review Board), as evidence suggests that trauma research is considered minimal risk [18, 19].

The overall sample consisted of 535 participants. Of this sample, 67.8% ($n$ = 363) identified as female, 31.4% ($n$ = 168) identified as male, 0.6% ($n$ = 3) identified as non-binary, and 0.2% ($n$ = 1) preferred not to report gender. The majority of the sample (68.4%; $n$ = 366) identified as Caucasian; 1.3% ($n$ = 7) identified as American Indian/Alaskan Native, 15.7% ($n$ = 84) identified as Asian, 2.3% ($n$ = 12) identified as African American, 1.3% ($n$ = 7) identified as Native Hawaiian/Pacific Islander, and 11.0% ($n$ = 59) identified as Other. Participants' ages ranged from 18 to 23, with an average age of 19.20 ($SD$ = 1.36). Before data analysis, one participant was excluded due to extensive missing data (all survey items left blank). Another participant was removed due to a response pattern characterized by high leverage on both independent variables and outlier status on the dependent variable. Because of this response pattern combined with the participant's nonsensical gender response, "11th dimension camera" (a phrase which appears nowhere on the internet), we opted to censor this participant.

### Measures

**Gender harassment.** Gender harassment was measured using the 20-item Gender Experiences Questionnaire (GEQ) [3]. The GEQ assesses five dimensions of gender harassment: Sexist Remarks (4 items), Sexually Crude/Offensive Behavior (5 items), Infantilization (3 items), Work/Family Policing (4 items), and Gender Policing (4 items). Initially developed to measure

women's experiences of workplace gender harassment in the past year [3], the GEQ was modified for this study in two ways. Rather than reporting on experiences in the workplace, participants were instructed to report on experiences of gender harassing behaviors during high school perpetrated by anyone associated with their high school, including classmates, coaches, teachers, school staff, and administrators. Three items from the Gender Policing subscale were also adapted to be more appropriate for male and gender-nonconforming participants. For example, an item on the original scale asked participants how frequently they were "made to feel like less of a woman because you had traditionally masculine interests." This item was altered to "made to feel like less of a man because you had traditionally feminine interests" for male participants and "made to feel like less of a person because your interests were gender-nonconforming" for gender-nonconforming participants.

Participants rated each item on a response scale ranging from 1 ("Never") to 5 ("Very Often"). Items from each subscale were then summed and averaged to create composite sub-scale scores ranging from 1 to 5, where higher scores represent more exposure to the respective dimension of gender harassment. All 20 GEQ items were summed and averaged to create a global composite GEQ score ranging from 1 to 5, where higher scores represent more exposure to total gender harassment. In prior research, the GEQ global and subscale scores have demonstrated excellent reliability ($\alpha$'s ranging from .78 to .93) and validity [3]. In this study, the overall scale demonstrated excellent reliability ($\alpha$ = .94). The reliability of GEQ items were roughly equivalent for both women and men. Among women, the overall GEQ scale ($\alpha$ = .95), Sexist Remarks subscale ($\alpha$ = .96), Sexually Crude/Offensive Behavior subscale ($\alpha$ = .91), Infantilization subscale ($\alpha$ = .92), Work/Family Policing subscale ($\alpha$ = .92), and Gender Policing subscale ($\alpha$ = .85) demonstrated excellent reliability. Among men, the overall GEQ scale ($\alpha$ = .92), Sexist Remarks subscale ($\alpha$ = .86), Sexually Crude/Offensive Behavior subscale ($\alpha$ = .85), Infantilization subscale ($\alpha$ = .89), Work/Family Policing subscale ($\alpha$ = .89), and Gender Policing subscale ($\alpha$ = .85) demonstrated similar reliability. For details on our confirmatory factor analysis of the GEQ subscales, please see S1 Table. The distribution of scores (*Skew* = 0.70, *Kurtosis* = 0.22) was within the acceptable ranges.

**Institutional betrayal.** Institutional betrayal was measured using the 12-item Institutional Betrayal Questionnaire (IBQ) [20]. The IBQ assesses participants' experiences with a range of institutional actions or inactions (e.g., "creating an environment in which sexual harassment seem common or normal"; "not taking proactive steps to prevent sexual harassment"; "making it difficult to report sexual harassment"). The 12-item scale consists of seven items included in the original version of the IBQ, as well as five additional items used in later research [20, 21]. When beginning the questionnaire, participants were provided with the following instructions: "Please consider your high school. (If you attended multiple high schools, please consider the high school at which you spent the most time enrolled.) How much do the following statements apply to the high school's attitudes and policies around sexual harassment?". Unlike prior research that used a dichotomous yes/no response format [20], IBQ items in this study were rated on a continuous response scale ranging from 0 to 3, where 0 corresponds to "Very False," 1 corresponds to "Somewhat False," 2 corresponds to "Somewhat True," and 3 corresponds to "Very True." All items were summed to create a total composite IBQ score ranging from 0 to 36, where higher scores represent more frequent institutional betrayal. In this study, the IBQ demonstrated excellent reliability ($\alpha$ = .94). The distribution of scores (*Skew* = 0.64, *Kurtosis* = -0.38) was within the acceptable ranges.

**Trauma symptoms.** Trauma symptoms were measured using the 40-item Trauma Symptom Checklist (TSC) [5]. The TSC assesses a diverse range of common posttraumatic symptoms, including headaches, memory problems, anxiety attacks, nightmares, sexual problems, and insomnia. Participants were instructed to rate how often they experienced each symptom

in the past month, with response options ranging from 0 ("Never") to 3 ("Often"). All 40 items were summed to create a total composite TSC score ranging from 0 to 120, where higher scores represent more frequent trauma symptoms. In prior research, the TSC has demonstrated excellent reliability ($\alpha$ = .90) and validity [5]. In this study, the TSC demonstrated excellent reliability ($\alpha$ = .93). The distribution of scores (*Skew* = 0.93, *Kurtosis* = 0.78) was within the acceptable ranges.

**Demographic information.** In a demographics questionnaire, participants were asked to report their age, gender, and race.

## Procedure

An online survey containing all study materials was created through Qualtrics. The link to the survey was distributed to participants in the Human Subjects Pool using SONA online software. After clicking a link that directed them to the Qualtrics survey, participants were required to undergo an informed consent process and indicate their consent to participate. Participants completed the survey on personal electronic devices, and they were provided with course credit as compensation. Upon completion of the survey, participants were presented with a debriefing form. All research procedures were approved by the institution's Office of Research Compliance (Institutional Review Board).

## Data analysis

Data were analyzed using *R* Version 3.5.2 and *R* packages *stats* (Version 3.5.2), *tidyverse* (Version 1.2.1), *psych* (Version 1.8.12), and *e1071* (Version 1.7–0.1) [22–25]. In order to test our hypotheses, we used linear regression analyses in which gender harassment, institutional betrayal, and gender harassment x institutional betrayal predicted current trauma symptoms, with participant age, gender, and race included as covariates. Continuous independent variables were standardized. The analysis plan was pre-registered on the Open Science Framework prior to any data observation (available at: https://osf.io/3ds8k).

**Missing data.** At the item level, 63 out of 38,520 items were missing (0.16%). One participant out of 535 was missing age (0.18%). At the questionnaire level, we conducted the Little MCAR test, and it was not significant. Therefore, we assume the data are missing completely at random. Missing data were imputed using single imputation through *R* package *Amelia* (Version 1.7.5) [26].

## Results

### Descriptive results

In the sample, 96.7% of women and 96.4% of men reported at least one instance of gender harassment in high school; only 12 women and 6 men reported no harassment experiences (i.e., scored the lowest possible score on the GEQ). Additionally, 87.3% of women and 76.7% of men reported at least one instance of institutional betrayal in high school; only 46 women and 39 men did not report institutional betrayal experiences (i.e., scored the lowest possible score on the IBQ). Participants had an average score of 2.25 (*SD* = 0.81) on the GEQ, 10.17 (*SD* = 8.67) on the IBQ, and 24.67 (*SD* = 17.07) on the TSC. Separate means and standard deviations for each gender category are reported in Table 1.

Independent t-tests (with equal variances assumed) were conducted to assess gender differences in TSC scores, IBQ scores, GEQ scores, and the five GEQ subscale scores. In order to correct for the number of statistical tests, a Bonferroni correction was implemented, resulting in an adjusted alpha threshold of 0.0063. Because of low cell sizes, gender non-conforming

**Table 1. Means and standard deviations by gender (N = 535).**

| Gender | n | Gender Harassment | | Institutional Betrayal | | Trauma Symptoms | |
|---|---|---|---|---|---|---|---|
| | | *M* | *SD* | *M* | *SD* | *M* | *SD* |
| 1. Women | 363 | 2.32 | 0.84 | 11.46 | 9.01 | 26.58 | 17.07 |
| 2. Men | 168 | 2.11 | 0.69 | 7.29 | 6.90 | 20.24 | 16.10 |
| 3. Non-Binary | 3 | 2.97 | 1.15 | 18.67 | 15.70 | 47.33 | 15.28 |
| 4. Prefer Not to Say | 1 | 1.20 | NA | 1.06 | NA | 10.00 | NA |

*M* and *SD* are used to represent mean and standard deviation, respectively.

($n$ = 3) participants were excluded from these analyses. Women had significantly higher scores than men on: the TSC, $t(529)$ = 4.05, $p < .001$; the IBQ, $t(529)$ = 5.32, $p < .001$; the overall GEQ, $t(529)$ = 2.84, $p = .005$; the sexist remarks GEQ subscale, $t(529)$ = 3.87, $p < .001$; the infantilization GEQ subscale, $t(529)$ = 2.78, $p = .006$; and the work/family policing GEQ subscale, $t(529)$ = 3.08, $p = .002$. Men and women's scores did not significantly differ on the gender policing GEQ subscale, $t(529)$ = 2.06, $p = .04$, or the sexually crude/offensive behavior GEQ subscale, $t(529)$ = -0.07, $p = .94$. Men and women's scores (and standard errors) on the GEQ subscales are displayed in Fig 1.

## Correlational results

Pearson's $r$ bivariate correlation coefficients were calculated between continuous variables (see Table 2 for full correlation matrix). Significant positive associations were found between gender harassment, institutional betrayal, and trauma symptoms, $p < .001$. There was also a significant positive association between age and institutional betrayal, $p < .01$. The association

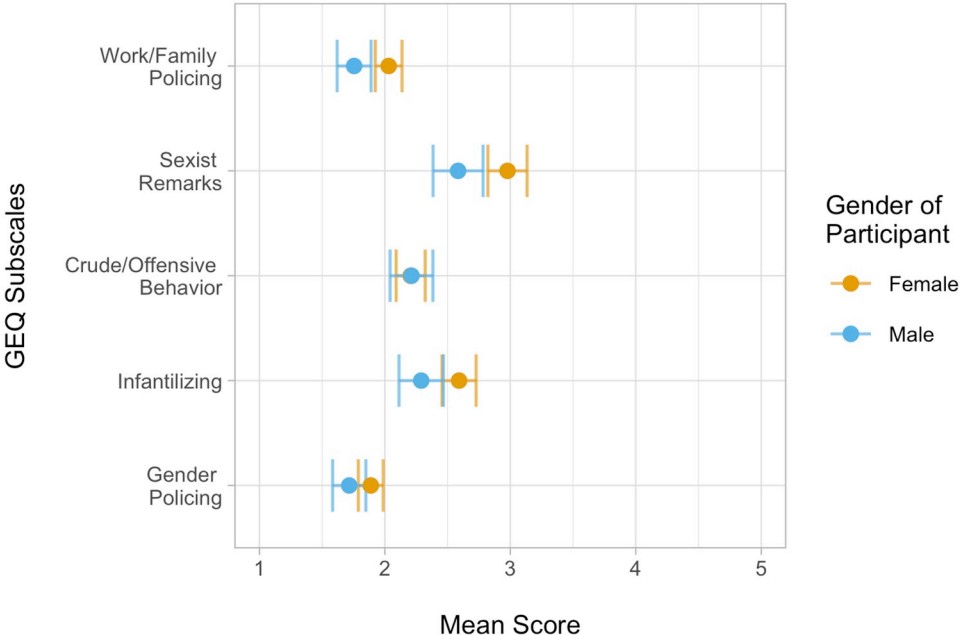

**Fig 1. Mean GEQ subscale scores by gender of participant (N = 535; 363 women, 168 men) with standard error bars.** Due to small cell size ($n$ = 3), we excluded non-binary participants from this figure. Figure created using *R* packages *ggplot2* (Version 3.1.0) and *colorblindr* (Version 0.1.0) [27, 28].

**Table 2. Means, standard deviations, and correlations of continuous variables (N = 535).**

| Variable | *M* | *SD* | 1 | 2 | 3 |
|---|---|---|---|---|---|
| 1. Age | 19.30 | 1.36 | | | |
| 2. Gender Harassment | 2.25 | 0.81 | .03<br>[-.05, .11] | | |
| 3. Institutional Betrayal | 10.17 | 8.67 | .12**<br>[.03, .20] | .45***<br>[.38, .51] | |
| 4. Trauma Symptoms | 24.67 | 17.07 | .01<br>[-.07, .10] | .39***<br>[.31, .46] | .36***<br>[.28, .43] |

between gender harassment scores on the GEQ and trauma symptom scores on the TSC are depicted in Fig 2. The association between institutional betrayal scores on the IBQ and trauma symptom scores on the TSC are depicted in Fig 3.

## Multiple regression models

In order to test our main hypotheses, we used multiple regression to predict trauma symptoms (see Table 3 for complete regression table). In the first step of the model, we included participant age, gender (with female as reference group), and race (with Caucasian as reference group). These demographic characteristics together significantly predicted trauma symptoms, $F(9, 525) = 3.61$, $p < .001$, $\Delta R^2 = .06$. In this model, being male was associated with lower trauma symptoms, $b = -6.69$, $t(525) = -4.24$, $p < .001$, and identifying as "Other" race was associated with higher trauma symptoms, $b = 5.00$, $t(525) = 2.11$, $p = .04$. In the second step of the

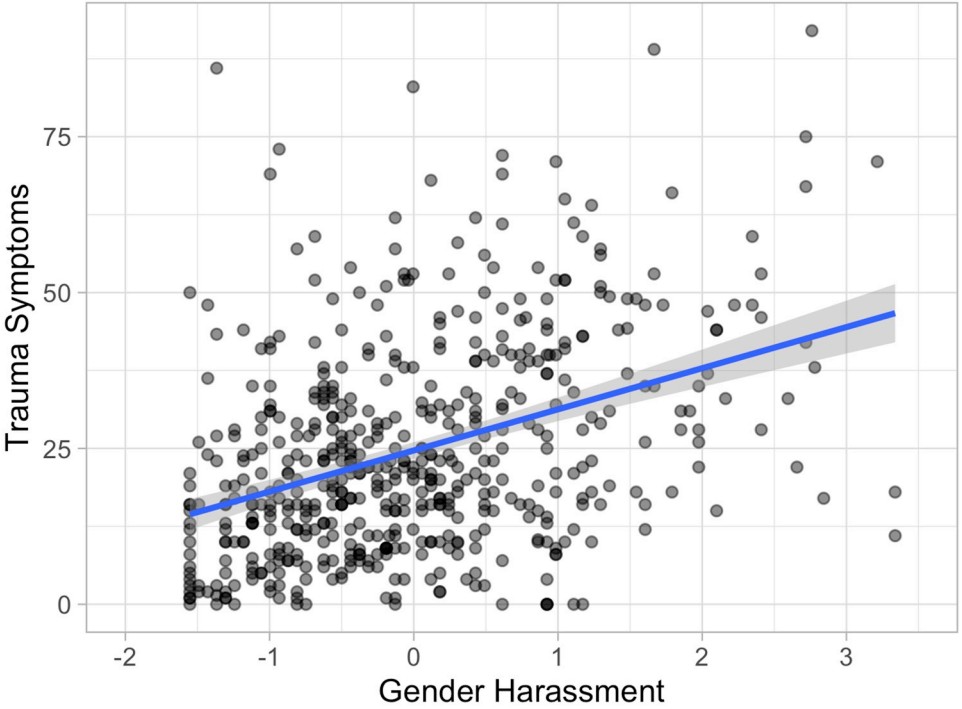

**Fig 2. Relationship between trauma-related symptoms and high school gender harassment.** Current trauma-related symptoms (raw TSC score) associated with self-reported high school gender harassment (standardized GEQ scores), plus line fit with "lm" method and 95% confidence interval. Figure created using *R* package *ggplot2* (Version 3.1.0) [28].

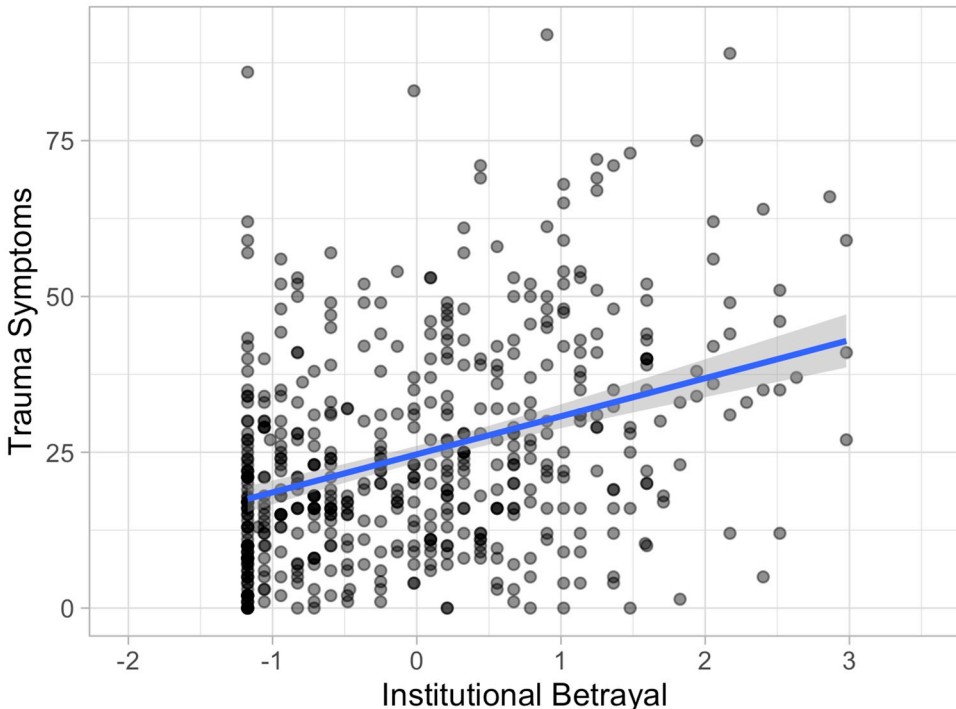

**Fig 3. Relationship between trauma-related symptoms and high school institutional betrayal.** Current trauma-related symptoms (raw TSC score) associated with self-reported high school institutional betrayal (standardized IBQ scores), plus line fit with "lm" method and 95% confidence interval. Figure created using *R* package *ggplot2* (Version 3.1.0) [28].

model, we included the main effects of gender harassment and institutional betrayal. A model comparison test indicated that the addition of these two predictors added significant predictive power to the model, $F(2, 523) = 51.53$, $p < .001$, $\Delta R^2 = .16$. In this step, both gender harassment, $b = 4.63$, $t(523) = 6.15$, $p < .001$, and institutional betrayal, $b = 3.60$, $t(523) = 4.70$, $p < .001$, were significant predictors of trauma symptoms. In the third step of the model, we included a gender harassment x institutional betrayal interaction term. A model comparison test indicated that the addition of this interaction term was not significant, $F(1, 522) = 2.10$, $p = .15$, $\Delta R^2 = .003$.

## Discussion

Our results suggest that both gender harassment and institutional betrayal are common experiences in high school and that females report significantly higher rates of both than do males. Confirming our first pre-registered hypothesis, our results indicate that high school gender harassment is associated with college trauma-related symptoms. Contrary to our second pre-registered hypothesis, high school institutional betrayal did not moderate the relationship between gender harassment and trauma-related symptoms. Our findings did, however, show that institutional betrayal is associated with trauma-related symptoms above and beyond the predictive power of age, gender, race, and gender harassment.

Our study replicates three important findings. First, we found a high prevalence of gender harassment in high school, even higher than rates reported by the AAUW [4]. Second, we found a substantial relationship between the experience of sex-based harassment and negative outcomes, which is well established in the literature [29]. Third, we found that males reported

**Table 3. Current trauma symptoms predicted by age, gender, race/ethnicity, gender harassment, institutional betrayal, and gender harassment x institutional betrayal (N = 535).**

| Predictor | b | b, 95% CI, [LL, UL] | Fit | Difference |
|---|---|---|---|---|
| Intercept | 15.56 | [-5.08, 36.21] | | |
| Age | 0.57 | [-0.51, 1.65] | | |
| American Indian/Alaskan Native | 4.63 | [-7.90, 17.16] | | |
| Asian | -3.05 | [-7.07, 0.96] | | |
| African American | 3.46 | [-6.17, 13.10] | | |
| Pacific Islander/Native Hawaiian | -1.89 | [-14.43, 10.64] | | |
| Other | 5.00* | [0.35, 9.65] | | |
| Male | -6.68*** | [-9.78, -3.59] | | |
| Non-Binary | 18.36 | [-0.78, 37.51] | | |
| Prefer Not to Say | -23.63 | [-57.01, 9.76] | | |
| | | | $R^2$ = .058** | |
| | | | 95% CI[.01,.09] | |
| Intercept | 27.04** | [7.93, 46.14] | | |
| Age | -0.10 | [-1.10, 0.90] | | |
| American Indian/Alaskan Native | 2.15 | [-9.35, 13.65] | | |
| Asian | -0.56 | [-4.28, 3.16] | | |
| African American | 3.10 | [-5.73, 11.92] | | |
| Pacific Islander/Native Hawaiian | 2.47 | [-9.05, 13.98] | | |
| Other | 4.86* | [0.60, 9.12] | | |
| Male | -3.48* | [-6.41, -0.55] | | |
| Non-Binary | 13.11 | [-4.45, 30.67] | | |
| Prefer Not to Say | -9.81 | [-40.50, 20.88] | | |
| Gender Harassment (GEQ) | 4.63*** | [3.15, 6.10] | | |
| Institutional Betrayal (IBQ) | 3.60*** | [2.09, 5.10] | | |
| | | | $R^2$ = .213** | $\Delta R^2$ = .155** |
| | | | 95% CI[.14,.26] | 95% CI[.10, .21] |
| Intercept | 26.96** | [7.87, 46.05] | | |
| Age | -0.12 | [-1.12, 0.88] | | |
| American Indian/Alaskan Native | 3.13 | [-8.43, 14.70] | | |
| Asian | -0.47 | [-4.18, 3.25] | | |
| African American | 3.21 | [-5.61, 12.03] | | |
| Pacific Islander/Native Hawaiian | 2.47 | [-9.03, 13.98] | | |
| Other | 4.93* | [0.67, 9.19] | | |
| Male | -3.50* | [-6.43, -0.58] | | |
| Non-Binary | 11.55 | [-6.11, 29.22] | | |
| Prefer Not to Say | -11.02 | [-41.73, 19.68] | | |
| Gender Harassment (GEQ) | 4.49*** | [3.00, 5.97] | | |
| Institutional Betrayal (IBQ) | 3.45*** | [1.93, 4.96] | | |
| GEQ x IBQ | 0.97 | [-0.35, 2.30] | | |
| | | | $R^2$ = .216** | $\Delta R^2$ = .003 |
| | | | 95% CI[.14,.26] | 95% CI[-.01, .01] |

experiencing less harassment than females, again mirroring a well-established pattern of results [30].

Our study bolsters the reliability and extends the utility of two recently developed measures. We successfully adapted and deployed the Gender Experiences Questionnaire (GEQ) and the

Institutional Betrayal Questionnaire (IBQ) for novel uses, while maintaining strong reliability [3, 20]. We adapted the GEQ for use with all genders, and we adapted the IBQ to capture institutional betrayal at the climate level. Our study also boasts the first use of both questionnaires to assess high school experiences.

Two innovative findings stand out as the most important contributions of this study. First, identifying the relationship between high school gender harassment and college trauma-related symptoms provides support for the continued investigation of gender harassment as an impediment to healthy adolescent development. This finding suggests that gender harassment merits increased attention in high schools. Second, high school institutional betrayal's association with college trauma-related symptoms above and beyond covariates plus gender harassment highlights the responsibility of schools to grapple with the potential harm caused by their indifference to harassment in their hallways. This finding extends the foundational research on institutional betrayal by (1) focusing on gender harassment and institutional betrayal in adolescence and (2) attempting to capture institutional betrayal at the climate level.

Two possible explanations may account for the finding that institutional betrayal did not moderate the relationship between gender harassment and trauma-related symptoms. First, the main effect of institutional betrayal may indicate that a high school's indifference to gender harassment may hurt students regardless of whether the student experiences gender harassment directly. In other words, it is possible that a climate of tolerating gender harassment may harm students. Second, the IBQ only captures the negative end of a high school's possible response to harassment, thus missing the potential benefit of a high school demonstrating institutional courage [31]. Expanding the scale to capture institutional courage might increase the likelihood of identifying a meaningful interaction.

## Limitations

Significant limitations constrain the conclusions drawn from this study. First and most importantly, we make no causal claims from these data because they are cross-sectional. Second, we did not account for possible confounders like childhood trauma and other types of high school sexual misconduct, which may reduce the share of trauma-related symptoms that are explained by gender harassment and institutional betrayal [32]. Third, though participants were blinded to the study's contents and therefore unlikely to be biased by self-selection, we did draw from a sample of college students, which may introduce bias and severely limit generalizability [17, 33]. Fourth, our survey lacked attention checks, which may have inflated the reliability and degraded the validity of the data [34]. Finally, we used retrospective report to measure high school gender harassment and institutional betrayal. While evidence suggests that retrospective report of child abuse holds steady over time, retrospective report shares some of the drawbacks of cross-sectional designs, i.e., the inability to describe the natural history of a phenomenon [35].

## Implications

This study advances several important ideas with implications for educators. Gender harassment in high school may interfere with the important work of adolescence. Gender harassment and institutional betrayal may turn high school into a hostile work environment. Schools may attempt to address these issues by exercising institutional courage. Institutional courage includes strong, transparent policy that complies with criminal laws and civil codes, voluntary accountability and the willingness to apologize, sensitive responses to disclosure, self-study and regular anonymous surveys, and the commitment of resources to all of the above [36]. Students' own suggestions parallel these recommendations [4].

## Future directions

In the near future, we plan to rerun this study with the addition of the AAUW sex-based harassment items, a measure of homophobic teasing, a measure of previous trauma exposure, and attention checks added in to replicate, extend, and strengthen these findings. More broadly, next steps should include longitudinal study designs with prospective recruitment to better describe the natural history of the phenomena. Additional next steps should include a social ecological approach to investigating the complex system in which gender harassment and institutional betrayal occur, including a focus on the student's relationship to and expectations of the high school; this approach could provide important insight for designing interventions. Finally, the development and deployment of an institutional courage-based intervention for schools could support causal claims and healthy adolescent development.

## Supporting information

**S1 Table. Factor loadings on subscales of the Gender Experiences Questionnaire (Leskinen & Cortina, 2014; N = 535).** For our factor analysis, we tested a 5-factor model, using maximum likelihood estimation with an oblique promax rotation, in order to be consistent with the original procedure used by Leskinen and Cortina (2014). In interpreting our model, we referred to benchmarks described in the literature to evaluate the fit. These benchmarks state that the fit is suitable when SRMSR is .10 or less, RMSEA is .08 or less, and TLI is .95 or higher (Williams et al., 2012). Results indicated that the SRMSR was excellent (.04), but the RMSEA value (.116) was higher than recommended at (90% CI[.109, .124]), and the TLI was lower than desired (.85). However, the low TLI value and high RMSEA value were consistent with the initial results found in Leskinen and Cortina (2014). In order to address this issue in the initial scale development, Leskinen and Cortina (2014) adjusted their fit in order to allow residuals for certain items to correlate with one another; this improved the model fit. Results of the factor analysis were remarkably consistent with result from Leskinen and Cortina (2014), with one exception. Item 17 ("Referred to the workplace as a 'man's space' (e.g., women do not belong here)") loaded onto work/family policing, instead of gender policing. (PDF)

## Acknowledgments

The authors thank Nicholas B. Allen, Ph.D., Melissa L. Barnes, M.S., Lauren E. Kahn, Ph.D., and Alec M. Smidt, M.S., for their contributions to this project.

## Author Contributions

**Conceptualization:** Monika N. Lind, Alexis A. Adams-Clark, Jennifer J. Freyd.

**Formal analysis:** Monika N. Lind.

**Investigation:** Monika N. Lind.

**Methodology:** Monika N. Lind, Alexis A. Adams-Clark.

**Project administration:** Monika N. Lind.

**Resources:** Jennifer J. Freyd.

**Supervision:** Jennifer J. Freyd.

**Visualization:** Monika N. Lind, Alexis A. Adams-Clark.

**Writing – original draft:** Monika N. Lind, Alexis A. Adams-Clark.

**Writing – review & editing:** Monika N. Lind, Alexis A. Adams-Clark, Jennifer J. Freyd.

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
