## [Decision Letter · Decision Letter 0]

5 May 2020

PONE-D-20-00802

Isn’t high school bad enough already? Investigating the rates and correlates of gender harassment and institutional betrayal in high school

PLOS ONE

Dear Ms. Lind,

Thank you for submitting your manuscript to PLOS ONE. And please sorry for taking much longer than we all expected. After careful consideration, we feel that your manuscript has merit but does not fully meet PLOS ONE’s publication criteria as it currently stands. Therefore, we invite you to submit a revised version of the manuscript that addresses the points raised during the review process (please see below).

We would appreciate receiving your revised manuscript by Jun 19 2020 11:59PM. To enhance the reproducibility of your results, we recommend that if applicable you deposit your laboratory protocols in protocols.io, where a protocol can be assigned its own identifier (DOI) such that it can be cited independently in the future. For instructions see: http://journals.plos.org/plosone/s/submission-guidelines#loc-laboratory-protocols

We look forward to receiving your revised manuscript.

Kind regards,

Juan Cristobal Castro-Alonso, Ph.D.

Academic Editor

PLOS ONE

Journal Requirements:

1. Thank you for providing information about financial disclosure and competing interests. Please provide more specific information about competing interests. For further instructions, please see https://journals.plos.org/plosone/s/competing-interests

Reviewers' comments:

Reviewer's Responses to Questions

**Comments to the Author**

1. Is the manuscript technically sound, and do the data support the conclusions?

Reviewer #1: Yes

Reviewer #2: Yes

2. Has the statistical analysis been performed appropriately and rigorously? 

Reviewer #1: Yes

Reviewer #2: Yes

3. Have the authors made all data underlying the findings in their manuscript fully available?

Reviewer #1: Yes

Reviewer #2: Yes

4. Is the manuscript presented in an intelligible fashion and written in standard English?

Reviewer #1: Yes

Reviewer #2: Yes

5. Review Comments to the Author

Reviewer #1: Thank you for the opportunity to review this interesting article. It reports on the rates of gender harassment and institutional betrayal in high school and their association with traumatic symptoms later on in college. The study also looks at the moderating effect of institutional betrayal on the association between gender harassment and traumatic symptoms.

I believe this is a well-written and strong piece of research that contributes to a relevant and understudied field. While the study has some important limitations, the authors identify and discuss them appropriately. I, therefore, recommend accepting this manuscript once the authors have addressed the following issues:

Title: I believe the authors should include the outcome of the study (i.e. traumatic symptoms) in the title, to summarize the focus of the article more explicitly.

HLM: The authors need to provide more information regarding the hierarchical linear models (HLM) fitted. More specifically, the nested structure of the data that warrants the use of this modeling approach is not clearly explained (e.g. is it students nested within high schools, students nested within undergraduate courses).

The specification of the models should also be explained more clearly: How many levels are considered in the model? Which levels? What type of HLM (e.g. a random intercept model) is used? And, what are their distributional assumptions?

I suggest to provide the corresponding equations.

Other goodness of fit indicators that are commonly used in HLM could also be reported.

Also, Table 3 should show the results from the random part on the models.

The R package used to fit HLM was not reported either. I recommend reviewing one of the following HLM textbooks for the standard of reporting HLM results:

Hox, J. (2002). Multilevel analysis. Techniques and applications. Mahwah, NJ.: Lawrence Erlbaum.

Snijders, T. A. B., & Bosker, R. (2004). Multilevel analysis: an introduction to basic and advanced multilevel modeling. London: Sage.

Raudenbush, S. W. & Bryk, A. S. (2002). Hierarchical Linear Models: Applications and Data Analysis Methods.

Finch, W. H. et al. (2017). Multilevel Modeling Using R.

Ethics: While I understand that participants were not informed about the content of the study to avoid biased self-selection and that the study was approved by their institutional ethics’ committee, I believe it would be important to reflect on and discuss the inclusion of trigger warnings in future research and the risks of double victimization.

Results: I believe that the sections Correlational results and Simple regression models are somewhat redundant as they both report on bivariate associations. The authors should focus more on reporting the HLM results.

In pages 7-8, the authors state "Rather than reporting on experiences in the workplace, participants were instructed to report on experiences of gender harassing behaviors during high school by anyone associated with their high school, including classmates, coaches, teachers, school staff, and administrators." Should it also say "and themselves"?

In page 9, it says "All items were summed to create a total composite IBQ score ranging from 0 to 36, where higher scores represent more frequent institutional betrayal." If this was the case and there were missing data in some of the IBQ items, those cases would obtain lower scores. I believe averaging scores would be a better approach.

As the authors suggest, the items of GEQ questionnaire differs somewhat by gender. I would suggest running reliability analysis separately to check that the internal consistency is similar among groups.

Also, considering that some of the scales used were modified from their original versions and that they now refer to a different setting (high school instead of work place for GEQ), I would suggest running factor analysis and reporting on the factor loadings of the items in an appendix.

Reviewer #2: This paper investigates an important topic. It is a competently conducted piece of research, with a large sample, and the writing is of exemplary clarity.

The findings serve to confirm existing views, rather than providing any great new insight. They replicate three previous findings, and bolster two existing measures. Beyond this, they reach a conclusion that might also be arrived at by informal observation, common sense, and a minimum of ethical awareness: that gender harassment is endemic, and that institutions should do their best to mitigate the harm which it causes. However, replication is a useful process, and the confirmation of an understanding which might appear obvious offers a valuable riposte to those who for personal or ideological reasons would like to minimise the significance of gender harassment. Consequently I believe that the paper is worthy of publication.

I would have liked to have seen a deeper consideration of the relationship between adolescents and high schools. The concept of institutional betrayal has political implications, in as much as it makes the case, or assumes, that it is the role of the institution to intervene in social interactions which are often not illegal, and which do not threaten in an immediate way the viability of the institution. This position has much to recommend it, but it is not universally held, nor are its limits uncontested. Indeed, I would argue that adolescents who suffer harassment are caught up a complex web of action and reaction involving both the defence of and resistance to a dominant gender discourse, and the actions of institutions seeking to navigate a path within this contested domain. It is the complexity of these interactions which lead to the stability of harassment, and to a tendency to institutional paralysis. Within this context, it may be that harassed individuals have very low expectations of the ability of the institution to act to protect them, perhaps with good reason, or they may never even have considered if it is the job of the institution to carry out this function. If so, this could perhaps be of relevance to your second hypothesis, as institutional betrayal may be accepted as the norm. I would be interested to see how your line of research could start to shed light on the wide range of social structures and interpersonal interactions which I believe lie behind institutional betrayal.

6. PLOS authors have the option to publish the peer review history of their article (what does this mean?). If published, this will include your full peer review and any attached files.

Reviewer #1: No

Reviewer #2: No

---

## [Author Response · Author response to Decision Letter 0]

2 Jul 2020

Comment: Title: I believe the authors should include the outcome of the study (i.e. traumatic symptoms) in the title, to summarize the focus of the article more explicitly. 

Response: This is an excellent suggestion, which we have implemented.

Comment: The authors need to provide more information regarding the hierarchical linear models (HLM) fitted. More specifically, the nested structure of the data that warrants the use of this modeling approach is not clearly explained (e.g. is it students nested within high schools, students nested within undergraduate courses).

Response: The reviewer points out an important weakness in the reporting of our results. The term, “hierarchical,” lacks clarity, and many readers would likely have the same interpretation that the reviewer had. In this case, what we meant by “hierarchical regression” was a statistical test to show if variables of interest explain a statistically significant amount of variance in the dependent variable after accounting for all other variables. In other words, we did not use levels to nest our data. Unfortunately, because of the wide range of high schools that university students come from, it is outside the scope of the project to nest individual students within their respective schools in an MLM approach and investigate predictors at the school level. This is a limitation (and potential future direction) of this research. We have revised the text to clarify our approach.

Comment: While I understand that participants were not informed about the content of the study to avoid biased self-selection and that the study was approved by their institutional ethics’ committee, I believe it would be important to reflect on and discuss the inclusion of trigger warnings in future research and the risks of double victimization.

Response: The reviewer rightly suggests that this research needs to be conducted with great care for participants. We have added a reference from our research group in the Procedure to provide evidence that trauma-related poses minimal risk to participants.

Comment: I believe that the sections Correlational results and Simple regression models are somewhat redundant as they both report on bivariate associations. The authors should focus more on reporting the HLM results.

Response: The reviewer helpfully identifies an overlooked redundancy. We have condensed the correlational and simple regression results to include only the correlation table and the key bivariate association figures.

Comment: In pages 7-8, the authors state "Rather than reporting on experiences in the workplace, participants were instructed to report on experiences of gender harassing behaviors during high school by anyone associated with their high school, including classmates, coaches, teachers, school staff, and administrators." Should it also say "and themselves"?

Response: We revised this sentence to improve its clarity.

Comment: In page 9, it says "All items were summed to create a total composite IBQ score ranging from 0 to 36, where higher scores represent more frequent institutional betrayal." If this was the case and there were missing data in some of the IBQ items, those cases would obtain lower scores. I believe averaging scores would be a better approach. 

Response: This is a reasonable concern, and we direct the reviewer to our Missing Data section on page 11, where we note the extremely small amount of missing data and our approach to imputation. When we scored the IBQ, data were already imputed so there were no items missing at that point.

Comment: As the authors suggest, the items of GEQ questionnaire differs somewhat by gender. I would suggest running reliability analysis separately to check that the internal consistency is similar among groups.

Response: This is an excellent suggestion, which we have added to the paper.

Comment: Also, considering that some of the scales used were modified from their original versions and that they now refer to a different setting (high school instead of work place for GEQ), I would suggest running factor analysis and reporting on the factor loadings of the items in an appendix.

Response: This is an excellent suggestion, which we have added to the paper. It’s especially valuable because it turns out that all but one item load very strongly onto the appropriate subscale factors -- a remarkable result for an adapted scale in a new population.

Comment: I would have liked to have seen a deeper consideration of the relationship between adolescents and high schools. [...] I would be interested to see how your line of research could start to shed light on the wide range of social structures and interpersonal interactions which I believe lie behind institutional betrayal.

Response: This is a terrific insight, and we would love to see this research as well. We have added content to the Future Directions section to reflect these suggestions.

---

## [Decision Letter · Decision Letter 1]

3 Aug 2020

Isn’t high school bad enough already? Rates of gender harassment and institutional betrayal in high school and their association with trauma-related symptoms

PONE-D-20-00802R1

Dear Dr. Lind,

We’re pleased to inform you that your manuscript has been judged scientifically suitable for publication and will be formally accepted for publication once it meets all outstanding technical requirements.

Kind regards,

Juan Cristobal Castro-Alonso, Ph.D.

Academic Editor

PLOS ONE

Additional Editor Comments (optional):

Reviewers' comments:

Reviewer's Responses to Questions

**Comments to the Author**

1. If the authors have adequately addressed your comments raised in a previous round of review and you feel that this manuscript is now acceptable for publication, you may indicate that here to bypass the “Comments to the Author” section, enter your conflict of interest statement in the “Confidential to Editor” section, and submit your "Accept" recommendation.

Reviewer #1: All comments have been addressed

2. Is the manuscript technically sound, and do the data support the conclusions?

Reviewer #1: Yes

3. Has the statistical analysis been performed appropriately and rigorously? 

Reviewer #1: Yes

4. Have the authors made all data underlying the findings in their manuscript fully available?

Reviewer #1: Yes

5. Is the manuscript presented in an intelligible fashion and written in standard English?

Reviewer #1: Yes

6. Review Comments to the Author

Reviewer #1: The authors have carefully considered and incorporated the comments made in the first revision. I, therefore, recommend publication of the article. It was a pleasure reading your manuscript.

7. PLOS authors have the option to publish the peer review history of their article (what does this mean?). If published, this will include your full peer review and any attached files.

Reviewer #1: No

---

## [Editor Report · Acceptance letter]

7 Aug 2020

PONE-D-20-00802R1 

Isn’t high school bad enough already? Rates of gender harassment and institutional betrayal in high school and their association with trauma-related symptoms 

Dear Dr. Lind:

I'm pleased to inform you that your manuscript has been deemed suitable for publication in PLOS ONE. Congratulations! Your manuscript is now with our production department. 

Kind regards, 

on behalf of

Dr. Juan Cristobal Castro-Alonso 

Academic Editor

PLOS ONE